# Multiphysics Modeling and Analysis of Sc-Doped AlN Thin Film Based Piezoelectric Micromachined Ultrasonic Transducer by Finite Element Method

**DOI:** 10.3390/mi14101942

**Published:** 2023-10-18

**Authors:** Xiaonan Liu, Qiaozhen Zhang, Mingzhu Chen, Yaqi Liu, Jianqiu Zhu, Jiye Yang, Feifei Wang, Yanxue Tang, Xiangyong Zhao

**Affiliations:** 1Shanghai Normal University, Shanghai 200234, China; 1000511407@smail.shnu.edu.cn (X.L.); chenmz_0126@163.com (M.C.); 19351373333@163.com (Y.L.); zhujianqiu@sinap.ac.cn (J.Z.); yangjiye1919@163.com (J.Y.); ffwang@shnu.edu.cn (F.W.); yanxuetang@shnu.edu.cn (Y.T.); 2Key Laboratory of Optoelectronic Material and Device, Department of Physics, Shanghai Normal University, Shanghai 200234, China

**Keywords:** PMUT, Sc-doped AlN, multiphysics modeling, finite element method

## Abstract

This paper presents a Piezoelectric micromechanical ultrasonic transducer (PMUT) based on a Pt/ScAlN/Mo/SiO_2_/Si/SiO_2_/Si multilayer structure with a circular suspension film of scandium doped aluminum nitride (ScAlN). Multiphysics modeling using the finite element method and analysis of the effect of different Sc doping concentrations on the resonant frequency, the effective electromechanical coupling coefficient ($k_{eff}^{2}$) and the station sensitivity of the PMUT cell are performed. The calculation results show that the resonant frequency of the ScAlN-based PMUT can be above 20 MHz and its $k_{eff}^{2}$ monotonically rise with the increasing doping concentrations in ScAlN. In comparison to the pure AlN thin film-based PMUT, the static receiving sensitivity of the PMUT based on ScAlN thin film with 35% Sc doping concentration is up to 1.61 mV/kPa. Meanwhile, the static transmitting sensitivity of the PMUT is improved by 152.95 pm/V. Furthermore, the relative pulse-echo sensitivity level of the 2 × 2 PMUT array based on the Sc doping concentration of 35% AlN film is improved by 16 dB compared with that of the cell with the same Sc concentration. The investigation results demonstrate that the performance of PMUT on the proposed structure can be tunable and enhanced by a reasonable choice of the Sc doping concentration in ScAlN films and structure optimization, which provides important guidelines for the design of PMUT for practical applications.

## 1. Introduction

Ultrasonic transducers have been widely used in medical imaging, nondestructive testing, rangefinders, gesture recognition, fingerprint systems, etc. [1,2,3,4]. A well-established technology in the medical imaging market is the conventional ultrasound transducer, which uses body piezoelectric actuation. However, conventional ultrasound transducers are large, poorly acoustically matched and unsuitable for two-dimension (2D) arrays [5]. With the increasing advancement in microelectromechanical system (MEMS) technology, micromechanical ultrasonic transducers (MUTs) have gained more and more attention. The piezoelectric micromachined ultrasonic transducers (PMUTs) are driven by applying an excitation voltage between the top and bottom electrodes of the piezoelectric layer. Compared with capacitive micromechanical ultrasonic transducers (CMUTs), PMUTs have the advantages of a no dc bias, a higher frequency, improved impedance matching and an easy array formation [6,7]. The PMUTs are widely used in industrial and biomedical applications such as biomedical imaging and fingerprint sensing [8]. In addition, PMUTs also have the advantages of miniaturization, low power consumption and complementary metal-oxide semiconductor (CMOS) compatibility. In particular, PMUTs are easy to form arrays. Compared with single-element transducers, the convenience, high frame rate and dynamic focusing capabilities of array transducers make them more widely applicable for clinical diagnosis and treatment [9].

With the expanding interest in large PMUT arrays, there exists a need for practical and accurate simulation tools to aid in the design process. Smyth et al. [10] adopted a Green’s function approach to solve the axisymmetric vibration modes of the circular plate and verified the modes with a PMUT having a radius of 400 μm. Dangi et al. [11] reported a system-level approach based on analytical lumped models for PMUTs below 1 MHz. The traditional plate theory is only suitable for low-frequency PMUT elements with a large radius and pitch and requires a lot of computing power. On this basis, Lu et al. [12] established a 2D axismetrical model to simulate the electromechanical acoustic behavior of PMUT without substrate. Simplifying the 2D axisymmetric model of cylindrical PMUT saves computational memory and time. Considering the resonant frequencies and modal shapes changed by various geometric combinations, 2D numerical simulation cannot accurately analyze complex structural units or arrays. The design of PMUT involves acoustic, solid mechanics, electrical and other coupling fields, and the traditional model is no longer applicable. In our previous work, we established a PMUT model based on (1 − x)Pb(Mg_1/3_Nb_2/3_)O_3−x_PbTiO_3_ (PMN-PT) using the finite element method (FEM), and analyzed the influence of the shape of the top electrode on the performance of PMUT [13]. It is demonstrated that FEM can be used to analyze and optimize PMUT cells and arrays with complex structures involving multi-field coupling. Therefore, the multiphysics coupled PMUT is simulated by FEM in this paper to solve and study the transceiver sensitivity and dynamic characteristics.

It is known that the performance of ultrasonic transducers is mostly determined by the used piezoelectric materials. Typically, the fabrication of ultrasound transducers calls for its piezoelectric materials possessing high piezoelectric coefficient, high effective electromechanical coupling coefficients ($k_{eff}^{2}$) and so on. Currently, the most commonly used piezoelectric materials in the medical ultrasonic transducer market are lead zirconate titanate (PZT) and aluminum nitride (AlN) [14]. The PZT has high dielectric constant, however, its preparation process contains lead, so it is gradually being replaced in today’s environmental protection. PZT also has the disadvantages of high processing temperature, low deposition rate and incompatibility with COMS. Compared to PZT, the lead-free piezoelectric material AlN has the characteristics of low cost, a wide source of raw materials and compatibility with COMS, but also, weak piezoelectricity and small $k_{eff}^{2}$ [15,16]. The key material properties of PZT and AlN are shown in Table 1. Shelton et al. [17] studied PMUT based on AlN thin films, and the results showed that the $k_{eff}^{2}$ was 0.056%. Akiyama et al. [18] found that adding scandium (Sc) to AlN can improve its piezoelectric properties. Yuan et al. [19] found that the piezoelectric constant of Sc_0.41_Al_0.59_N film material can reach 31.6 pC/N. Many studies have shown that while maintaining the advantages of AlN, the piezoelectric response and $k_{eff}^{2}$ of ScAlN piezoelectric films have been significantly improved, which provides a broad prospect for the application of ScAlN piezoelectric films in PMUT devices. However, current research mainly focuses on the material properties of ScAlN, and the performance of the devices based on AlN films with different Sc doping concentrations are still insufficient. Therefore, in this paper, Sc doped AlN film is selected as the piezoelectric layer material, and the multi-physics coupling model of the ScAlN thin film-based PMUT is established and analyzed.

This paper proposed a PMUT based on a Pt/ScAlN/Mo/SiO_2_/Si/SiO_2_/Si multilayer structure with a circular suspension film. Multiphysics modeling using the FEM software COMSOL Multiphysics 5.5 and analysis of the effect of different Sc doping concentrations on the resonant frequency, the $k_{eff}^{2}$ and the station sensitivity of the PMUT cell are performed. It is found that when the Sc doping concentration is 35%, the static receiving sensitivity of the PMUT cell reaches the peak value of 1.61 mV/kPa. Based on this, a 3D model of the ScAlN-based thin film PMUT array was established, and the acoustic field characteristics of the array were calculated and simulated. It is found that the resonant frequency of the array based on the same Sc doping concentration is higher than that of the cell, and the relative pulse sensitivity level is significantly improved.

## 2. PMUT Modeling and Simulation

### 2.1. FEM Modeling of PMUT

The proposed 3D finite element model of a single PMUT element was created using COMSOL Multiphysics 5.5. Based on the bending mode, the PMUT consists of a suspended membrane and its supporting structure. Figure 1a shows a cross-section schematic of the designed PMUT with the Pt/ScAlN/Mo/SiO_2_/Si/SiO_2_/Si structure. From top to bottom: the top electrode Pt, the piezoelectric layer material ScAlN, the bottom electrode Mo, the insulating layer SiO_2_, the device Si layer, the SiO_2_ barrier layer and the Si support layer. In order to improve the sensitivity, the top electrode Pt is only partially covered with a piezoelectric layer [20]. The SiO_2_ layer above the top layer of silicon acts as an insulating and protective layer. The SOI wafer is isolated from the supporting silicon substrate by a SiO_2_ barrier layer, thus reducing parasitic capacitance and power consumption. Figure 1b shows the 3D finite element model (circular top electrode and circular bottom cavity) based on the ScAlN thin film PMUT. Since the setup of the circular top electrode and the circular bottom cavity is convenient for device preparation, the PMUT with the circular top electrode and the circular bottom cavity is chosen for the study in this paper. As for a PMUT with a ScAlN film thickness of 0.78 μm and a target resonance frequency around 20 MHz, its corresponding diameter should be ~50 μm according to the plate theory [21]. PMUTs typically have high aspect ratios, and the size of PMUTs needs to be reduced to tens of microns in order to meet the resolution requirements of ultrasonic imaging [8]. The geometric parameters of the 3D finite element model of the PMUT cell in this paper are shown in Table 2, and the overall size of the PMUT model is 60 µm × 60 µm. One point to note is that the neutral axis should be outside the piezoelectric layer to ensure that the device has good sensitivity [22]. The material parameters of ScAlN, Si and SiO_2_ are taken from refs [23,24].

As for the physics-field modeling, the built-in electrostatic module, and the acoustic module as well as the multi-physics module are employed. In the solid mechanics module, the bottom surface of the PMUT model is defined as a fixed constrained boundary condition. In the electrostatic module, the terminal boundary conditions are set and defined by the top electrode, and the ground boundary conditions are set and defined by the bottom electrode. Figure 2 shows the meshing grid elements in each domain, and the maximum size of the grid cells in a given material is specified as one-fifth of the wave-length in order to accurately resolve the stress waves in the solid domain. The maximum size of the grid cells in a given material is specified as one-fifth of the wavelength in order to accurately resolve the stress waves in the solid domain. Unlike the 3D model with a tetrahedral mesh, a thin plate requires at least three solid cells in the thickness direction. A 2 × 2 PMUT array was constructed on the basis of PMUT element. According to the acoustic imaging application, the spacing of the array elements usually needs to be between half to one wavelength (λ) of the ultrasonic waves in the propagating medium [25]. In the array design, a half-wavelength spacing of 70 µm was chosen to avoid imaging artifacts and the generation of side flaps in the beam direction map.

### 2.2. Evaluation of PMUT Properties

The most affected index in ultrasonic transducers is the information of the center frequency, sensitivity, directivity, and bandwidth of the transducer etc. In this paper, the resonant frequency and modal vibration pattern of PMUT are calculated by eigenfrequency analysis. The response near the eigenfrequency is simulated by frequency domain analysis to obtain the conductivity characteristics. In order to assess the electromechanical coupling without considering the geometric shape and excitation mode of PMUT, the $k_{eff}^{2}$ can be defined as [26]:(1)$$k_{eff}^{2}=1-{\left(\frac{f_{r}}{f_{a}}\right)}^{2}$$
where $f_{r}$ is the resonance frequency and $f_{a}$ is the anti-resonance frequency.

In addition, the static sensitivity of the PMUT is studied in this paper using stationary analysis. The static receiving sensitivity *s*_r_ can be used as an indicator of the receiving performance of the PMUT, while the static transmitting sensitivity *d*_s_ is used to evaluate the transmitting performance of the PMUT. When investigating the static transmitting sensitivity, a voltage of 10 V µm^−1^ is applied across the upper and lower electrodes. To study the static receiving sensitivity, a load pressure of 100 kPa is imposed on the surface of the film. The values of the displacement and voltage produced at the center of the membrane surface can be determined, respectively. Therefore, the ratio of displacement to voltage at the center point of the membrane surface may be used to calculate the transmitting sensitivity *d*_s_, and the ratio of voltage to pressure at the center point of the membrane surface can be used to estimate the static receiving sensitivity *s*_r_ [27]. The on-axis directivity of the PMUT can be explained via the directivity index (DI):(2)$$DI=10\log(2\pi r^{2}p^{2}/\rho cP_{tot})$$
where *r* is the distance to the center of PMUT, *p* is the far field pressure at distance *r*, *P*_tot_ is the total radiate power, *ρ* and *c* are the density and sound speed of water, respectively.

The dynamic characteristics of the PMUT with added water and water perfectly matched layers were investigated by time domain analysis. When the top and bottom electrodes of the PMUT are charged, a transverse stress is generated in the piezoelectric layer due to the converse piezoelectric effect. The generated stress causes a bending moment that forces the membrane to deflect out of plane and emits an acoustic pressure wave. Similarly, due to the direct piezoelectric effect, the incident pressure wave causes membrane deformation and charge on the electrodes. Therefore, the PMUT can act as both a transmitter and a receiver. When a certain pulse signal is applied to the top electrode, this signal is transmitted to the solid reflector. Then part of the signal is reflected and received by the PMUT [28].

When a particular pulse signal is introduced into the upper electrode, it is transmitted to the solid reflector. Subsequently, some of the signal is detected by the PMUT and some of the signal is reflected by the PMUT. The relative pulse-echo sensitivity level *M* is an indicator for evaluating the dynamic sensitivity performance of the PMUT. The relative pulse-echo sensitivity level is expressed in *M*. Its mathematical formulation can be written as follows:(3)$$M=20\mathrm{lg}\left(\frac{U_{max}}{U_{0}}\right)$$
where *U*_max_ is the maximum amplitude of the initial pulse-echo voltage reflected by a reflector, *U*_0_ is the peak excitation voltage associated with the PMUT.

The center frequency and the corresponding −6 dB bandwidth (BW) of the pulse-echo frequency spectrum are defined as follow:(4)$$f_{c}=\left(f_{h}+f_{l}\right)/2$$
(5)$$BW=\frac{f_{h}-f_{l}}{f_{c}}\times 100\%$$
where $f_{h}$ and $f_{l}$ are the frequencies correspond to a 6 dB drop from the maximum amplitude of the pulse-echo signal in the frequency domain [29].

## 3. Results and Discussion

### 3.1. Effect of Sc Concentration on PMUT Cells

In order to investigate the effect of the Sc doping concentration on the resonant frequency of PMUT, PMUT models based on ScAlN films with different Sc doping concentrations ranged from 0% to 40% were calculated. Figure 3 shows the variation in the PMUT resonant frequency with the Sc doping concentration. The resonant mode is shown as inset in Figure 3. As shown, the first-order resonant mode (0, 1) have concentrated energy in the medium and are superior to other modes [30], and thus, we focus on the study of this main mode in the following analysis. As seen from the figure, it can be seen that the resonant frequencies of the PMUT can be above 20 MHz for all the PMUT models based on different Sc doping concentrations. As the concentration of Sc doping increases, the resonant frequency of the PMUT cell decreases from 22.454 MHz (Sc = 0%) to 20.92 MHz (Sc = 40%).

The effect of the Sc doping concentration on the $k_{eff}^{2}$ of the PMUT cell is shown in Figure 4a. The results show that the $k_{eff}^{2}$ of the PMUT cells based on Sc-doped AlN films gradually augments with the increasing in the Sc concentration until the Sc doping concentration reaches 40%. When the Sc doping concentration is 40%, the $k_{eff}^{2}$ reaches 3.72 times higher than that without Sc doping (0.53%). This indicates that the Sc doping concentration can have a large effect on the $k_{eff}^{2}$ of PMUTs.

Figure 4b shows the variation in the static transmitting sensitivity and the static receiving sensitivity of the PMUT cell with the concentration. The static transmitting sensitivity shows an increasing trend with increasing the Sc doping concentration, and the variation is significant. When varying the Sc doping concentration from 0% to 40%, the static transmitting sensitivity increases to 285.06 pm/V. The results show that the static receiving sensitivity increases and then decreases as the Sc doping concentration increases, and the static receiver sensitivity is at the peak of 1.61 Mv/kPa when the Sc doping concentration is 35%.

### 3.2. Design and Optimizing of PMUT Array

#### 3.2.1. Static Analysis

The above study on the effect of the Sc doping concentration on the performance of PMUT cells show that the Sc doping concentration has a profound effect on the resonant frequency, $k_{eff}^{2}$ and sensitivity of PMUTs. Therefore, it is also important to investigate the effect of the Sc doping concentration on the performance of PMUT arrays for designing PMUTs. In Figure 3 and Figure 4, the static receiving sensitivity of the PMUT cell is at its peak at an Sc doping concentration of 35%, and the $k_{eff}^{2}$ and static transmitting sensitivity are also at large values at this time. Therefore, in this paper, a 2 × 2 PMUT array based on 35% Sc-doped AlN thin film is established on the basis of the PMUT cell, and the electroacoustic characteristics of the array are investigated. The established 2 × 2 PMUT array model is shown in Figure 2b. The transmission performance of the PMUT array in the acoustic field and the water loading in the acoustic field are simulated. Simulations with arrays with more cells were not performed due to computational volume limitations.

The resonant frequency of the PMUT is the factor that determines the resolution and penetration depth. The conductance of PMUT cell and 2 × 2 array based on AlN with an Sc doping concentration of 35% are shown in Figure 5. The resonant frequency of PMUT array is 21.412 MHz, which is higher than the resonant frequency of PMUT cell with the same concentration, and the $k_{eff}^{2}$ is 3.28%.

The directivity and on-axis pressure of the PMUT array were investigated and compared with that of the cell. Figure 6 and Figure 7 show the polar coordinate directivity and axial pressure for PMUT cells and 2 × 2 arrays. The polar plots depicted in Figure 6 and Figure 7 illustrate the radiation patterns within the yz-plane. It is evident that the proposed PMUT array exhibit greater directivity along the *z*-axis compared to alternative directions. Compared with the PMUT cell, the side lobes of the PMUT array increases and the axial pressure increases. In addition, the decay of the sound pressure of the 2 × 2 PMUT array is slower than that of the PMUT cell for the same excitation source.

#### 3.2.2. Dynamic Analysis

The dynamic transmission and reception characteristics of PMUTs based on Sc-doped AlN films in water medium were investigated using time-domain analysis. Figure 8a shows a model of an underwater PMUT with a circular top electrode and a circular bottom cavity. The steel plate is placed on the central axis of the PMUT as a solid reflector. The solid mechanics module includes the PMUT and the solid reflector, while the piezoelectric material is classified under the electrostatic module. The pressure acoustics module comprise the water domain and the PML. In order to improve computational efficiency, this paper adopts the method of using a free triangular mesh partition to handle reflecting surfaces, water and the PML. When applying a sinusoidal pulse of the resonant frequency of PMUT for two cycles on the top electrode, the PMUT switch to receiving mode. The sound field change in water can be simulated, after the sound wave reaching the steel plate, a portion of the sound wave is reflected back and detected by the PMUT, thereby establishing an infinite cycle. Figure 8b illustrates the changes in underwater sound pressure. It is evident that the sound waves produced by the PMUT in the water medium exhibit a consistent pattern.

Figure 9a demonstrates the charge of the PMUT at a distance of 250 µm. The image within the orange dashed box is the first impulse response. From Equation (3), the relative pulse-echo sensitivity level of the PMUT is about −27 dB. In addition, the variation in the relative pulse-echo sensitivity level as a function of reflector position was investigated by hanging steel plates as reflectors at different distances away from the PMUT. As shown in Figure 9b, it is seen that the relative pulse-echo sensitivity level decreases as the distance between the reflector and the PMUT increases.

The frequency-amplitude spectrum of the pulse-echo was calculated using the fast Fourier transform (FFT) based on the pulse-echo response. Figure 10 shows the pulse-echo (black plot) and frequency response (red plot) of the cell and 2 × 2 PMUT array based on an Sc doping concentration of 35% AlN film underwater when the reflector is at 250 µm. In Figure 10a, the PMUT cell has a center frequency of 13.3 MHz and −6 dB upper and lower frequencies of 11.2 and 15.4 MHz, respectively, yielding a bandwidth of about 31.6%. In addition, the pulse-echo sensitivity of the 2 × 2 PMUT array model underwater was also investigated. Figure 10b shows the 2 × 2 PMUT array with a center frequency of 14.2 MHz, a bandwidth of about 28.2%, and −6 dB upper and lower frequencies of 12.2 MHz and 16.2 MHz, respectively. Table 3 compares the performance in terms of the relative pulse-echo sensitivity levels, center frequencies, and bandwidths for the ScAlN-based PMUT cell and the 2 × 2 array with the Sc doping concentration of 35%. As seen from Table 3, the center frequency of the PMUT 2 × 2 array is higher than the center frequency of the cell. There exists a slight decrease in the bandwidth of the PMUT array compared to the cell based on the same concentration. The relative pulse-echo sensitivity level of the 2 × 2 PMUT array based on the Sc-doped AlN film with 35% concentration is 16 dB higher than that of the cell with the same concentration.

## 4. Conclusions

In this paper, PMUT cells and PMUT arrays based on Sc-doped AlN films with different concentrations are modeled by the finite element method, and their electromechanical and acoustic performances are calculated and analyzed. First, the influence of the Sc concentration on electromechanical characteristic of PMUT cells is studied. It is shown that the resonant frequency of the ScAlN-based PMUT can be above 20 MHz and the Sc doping concentration gives rise to significant enhancement. The $k_{eff}^{2}$ of the PMUT is enhanced by seven times when the Sc doping concentration is 40% compared with that of the PMUT based on pure AlN film, and the static transmitting sensitivity increases to 285.06 pm/V when the Sc doping concentration changes from 0% to 40%.

Furthermore, the acoustic field properties of the PMUT cell and array were investigated. The relative pulse-echo sensitivity level of the 2 × 2 PMUT array based on the Sc doping concentration of 35% AlN film is improved by 16 dB compared to that of the cell with the same Sc concentration. The investigation results demonstrate that the ScAlN-based PMUT on the proposed structure enables high operating frequency and obvious enhancement in performance by the properly chosen level of the Sc doping concentration, which provides important design guidelines for the practical application in high-frequency medical ultrasound imaging. Furthermore, the proposed FEM model can also provide the possibility for design of high performance PMUT with arbitrary structures and materials like other nitride films [31,32].

## Figures and Tables

**Figure 1 micromachines-14-01942-f001:**
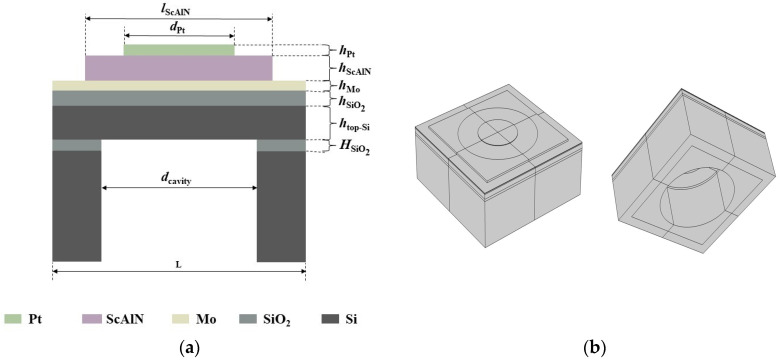
(**a**) Cross-sectional schematic of ScAlN-based PMUT; (**b**) 3D models of PMUT cell with circular top electrodes and circular bottom cavities for FEM simulation.

**Figure 2 micromachines-14-01942-f002:**
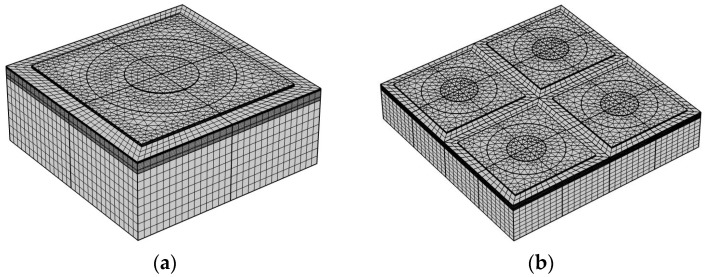
Mesh details of ScAlN-based PMUT model: (**a**) PMUT cell; (**b**) 2 × 2 PMUT array.

**Figure 3 micromachines-14-01942-f003:**
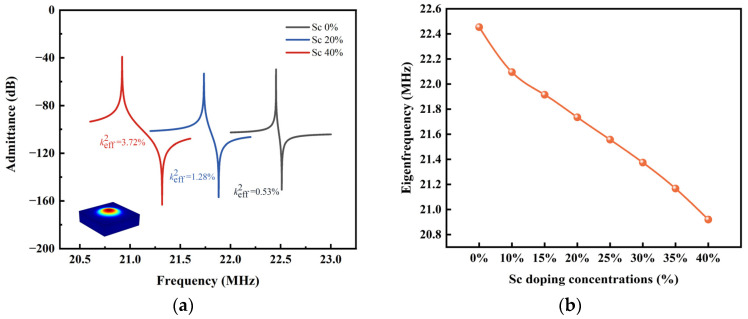
(**a**) The admittances of ScAlN-based PMUT cells; (**b**) Effect of Sc concentration on the resonant frequency of PMUT cells.

**Figure 4 micromachines-14-01942-f004:**
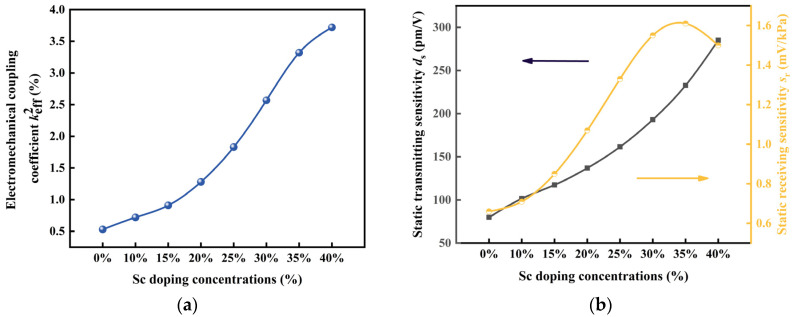
(**a**) Effect of the Sc concentration on the $k_{eff}^{2}$ of PMUT cells; (**b**) Effect of the Sc concentration on the static sensitivity of PMUT cells.

**Figure 5 micromachines-14-01942-f005:**
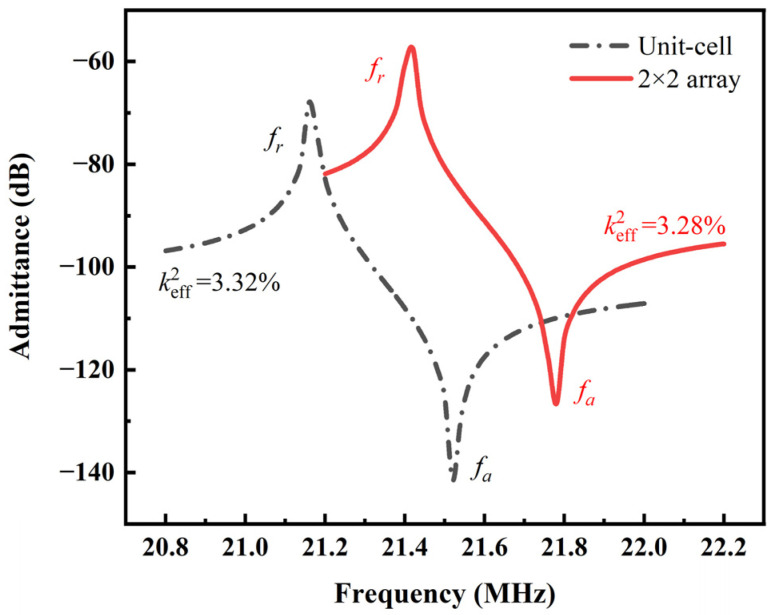
The calculated admittances of ScAlN-based PMUT cell and array.

**Figure 6 micromachines-14-01942-f006:**
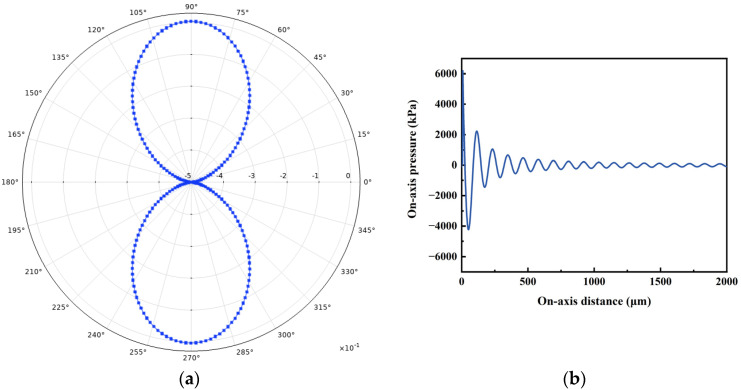
Directivity and on-axis pressure of PMUT cell: (**a**) Directivity; (**b**) On-axis pressure.

**Figure 7 micromachines-14-01942-f007:**
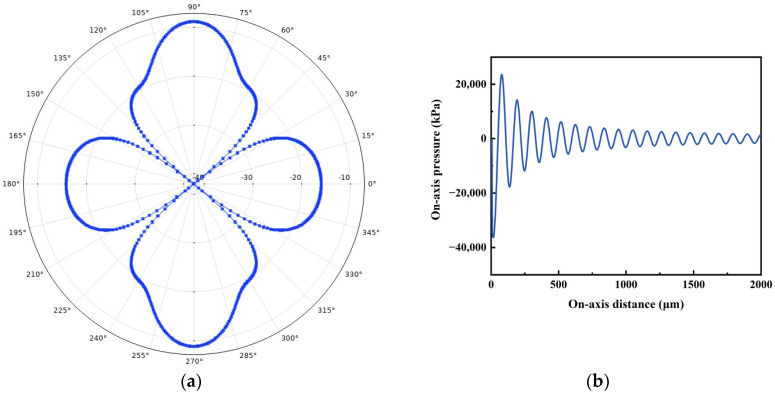
Directivity and on-axis pressure of PMUT 2 × 2 array: (**a**) Directivity; (**b**) On-axis pressure.

**Figure 8 micromachines-14-01942-f008:**
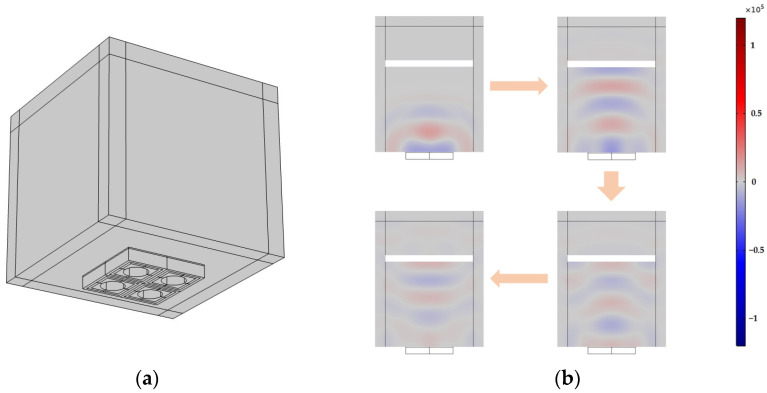
(**a**) The FEM model of PMUT array with an acoustic domain and an acoustic PML; (**b**) Variation in sound pressure at different times.

**Figure 9 micromachines-14-01942-f009:**
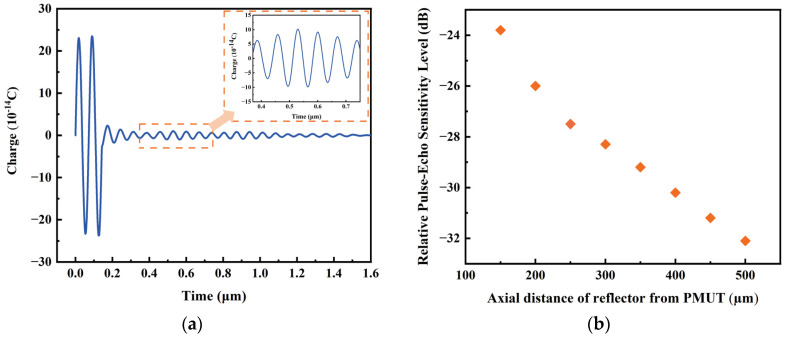
(**a**) Pulse-echo results with reflector at 250 µm; (**b**) Relative pulse-echo sensitivity level shifts with position of reflector.

**Figure 10 micromachines-14-01942-f010:**
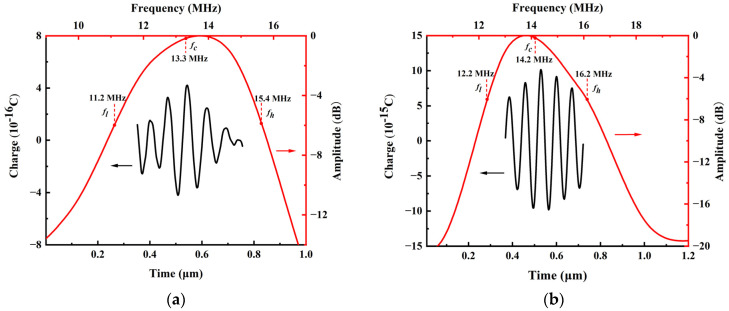
The pulse-echo (black plot) and the frequency response (red plot) of PMUT under water with reflector at 250 µm: (**a**) PMUT cell; (**b**) 2 × 2 PMUT array.

**Table 1 micromachines-14-01942-t001:** Key material properties of PZT and AlN.

Material Coefficient	PZT	AlN
$e_{31,f}\;\left(C\;m^{-1}\right)$	−8 to −12	−1.0
$d_{33}(pC\;N^{-1})$	200–400	~5.5
${\;\varepsilon }_{33}^{S}$	~1500	~10
$k_{33}$	~70%	~39%

**Table 2 micromachines-14-01942-t002:** Structural parameters of the ScAlN-based PMUT.

Parameter	Value
Thickness of top Pt	200 nm
Thickness of ScAlN	780 nm
Thickness of bottom Mo	190 nm
Thickness of SiO_2_	350 μm
Thickness of the top layer Si	2 μm
Thickness of SiO_2_	1 μm
Radius of top Pt	20 μm
Side length of ScAlN	50 μm
Radius of etching	20 μm

**Table 3 micromachines-14-01942-t003:** Center frequencies and bandwidths of cell and PMUT array models based ScAlN-based PMUT.

Parameter	35% Sc PMUT Cell	35% Sc 2 × 2 PMUT Array
Relative pulse-echo sensitivity level (dB)	−43	−27
Lower/upper −6 dB (MHz)	11.2/15.4	12.2/16.2
Center frequency (MHz)	13.3	14.2
Bandwidth (−6 dB)	31.6%	28.2%

## Data Availability

The data presented in this study are available on request from the corresponding author.

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
