# Peer review of "Multiphysics Modeling and Analysis of Sc-Doped AlN Thin Film Based Piezoelectric Micromachined Ultrasonic Transducer by Finite Element Method"

_micromachines, 2023, doi:10.3390/mi14101942_

Round 1
Reviewer 1 Report
In this paper, a finite element simulation of piezoelectric ultrasonic transducer based on ScAlN material is proposed. The investigation results demonstrate that the performance of PMUT on the proposed structure can be tunable and enhanced by a reasonable choice of the Sc doping concentration in ScAlN films and structure optimization, which provides important guideline for design of PMUT for practical applications. However, some aspects of the presentation of the article need to be improved. Here are some suggestions.
1. In the structural design, why choose Pt as the upper electrode instead of the material like Mo as the bottom electrode?
2. What is the role of the SiO2 layer above the top layer of silicon in the structure layer, please give a specific explanation.
3. For the relationship between the transverse size of the film and the wavelength, why choose 50um diameter, please give a detailed explanation.
There are some long English sentences in the article, for the sake of readability, please try to change the long sentences into phrases.
Reviewer 2 Report
In this paper, PMUT cells and PMUT arrays based on Sc-doped AlN films with dif-300 ferent concentrations are modeled by the finite element method, and their electromechan-301 ical and acoustic performances are calculated and analyzed. The paper is in good writing, here are some questions:
1. the authors only do modeling using FEM, is it possible for the author to do some experiments of PMUT for comparing?
2. Some labels in the Figures are not clear, e.g.: figure 8. the numbers are not clear.
3. Is it possible to replace the ScAlN film with a high-entropy nitride film? The author could give some short discussion about it in the revised manuscript, for example, the literatures "Entropy, 20, 2018: 624." " Thin Solid Films, 638, 2017: 383-388." should be referred.
Author Response
Please see the the attachment.

Reviewer 3 Report
The manuscript mainly presents FEM simulation results of AlScN pMUTs. The FEM simulation methodology, designed pMUT structure and tunable performance are well known. This work lacks novelty and significance.
Minor editing
Round 2
Reviewer 3 Report
1. It would be better if the authors can compare or comment the device performances with the state-of-the-art pMUTs made of AlN, AlScN, sol-gel/sputtered PZT, ceramic PZT, such as:
https://ieeexplore.ieee.org/abstract/document/7349139
https://www.sciencedirect.com/science/article/pii/S0924424722003041
https://ieeexplore.ieee.org/abstract/document/9493855
2. The authors presents the simulation of the pMUT array, but it is only a 2x2 array. For most of applications, pMUT array has large number of elements, such as:
https://www.nature.com/articles/s41378-022-00449-0
https://www.mdpi.com/2072-666X/13/6/962
Please consider and elaborate this section.
minor editing